# Evaluation of the Meet-URO Score in a Real-World Cohort of mRCC Patients Treated with First-Line TKIs

**DOI:** 10.3390/jcm14186385

**Published:** 2025-09-10

**Authors:** Faruk Recep Özalp, Ece Şahin Hafızoğlu, Ahmet Melih Arslan, Mehmet Çakmak, Sedat Biter, Rezan Berkay İzgör, Oktay Halit Aktepe, Ferhat Ekinci, Hüseyin Salih Semiz, İsmail Oğuz Kara, Mustafa Erman, Şuayib Yalçın, Aziz Karaoğlu

**Affiliations:** 1Department of Medical Oncology, Dokuz Eylul University Faculty of Medicine, Izmir 35340, Turkeyazizk1935@gmail.com (A.K.); 2Department of Medical Oncology, Celal Bayar University Faculty of Medicine, Manisa 45030, Turkey; 3Department of Medical Oncology, Bilkent City Hospital, Ankara 06800, Turkey; 4Department of Medical Oncology, Çukurova University Faculty of Medicine, Adana 01250, Turkey; 5Department of Internal Medicine, Dokuz Eylul University Faculty of Medicine, Izmir 35340, Turkey; 6Department of Medical Oncology, Hacettepe University Faculty of Medicine, Ankara 06230, Turkey

**Keywords:** metastatic renal cell carcinoma, tyrosine kinase inhibitor, Meet-URO score, IMDC, prognostic model

## Abstract

**Background:** The Meet-URO score combines the International Metastatic Renal Cell Carcinoma Database Consortium (IMDC) criteria with neutrophil-to-lymphocyte ratio (NLR) and bone metastasis status. Although developed in immune checkpoint inhibitor (ICI) cohorts, its performance among patients receiving first-line tyrosine kinase inhibitor (TKI) monotherapy is uncertain. **Methods:** We performed a multicenter, retrospective cohort study of 301 adults with histologically confirmed metastatic renal cell carcinoma (mRCC) treated with first-line TKI monotherapy (sunitinib, pazopanib, or cabozantinib) between 2008 and 2025 across five tertiary centers in Turkey. The primary endpoint was overall survival (OS). Meet-URO was calculated at treatment start and analyzed as prespecified risk strata (0–4, 5–8, 9). Kaplan–Meier estimates, Cox models, and Harrell’s C-index assessed discrimination, with bootstrapped 95% CIs. **Results:** Median follow-up was 40 months; median OS (mOS) was 25 months (95% CI, 21–29) and median progression-free survival was 10 months (95% CI, 8–12). Meet-URO stratified OS: 41 months for scores 0–4, 21 months for 5–8, and 7 months for 9 (log-rank *p* < 0.001). In multivariable analysis, Meet-URO remained independently prognostic (HR 1.73 for 5–8 vs. 0–4; HR 3.57 for 9 vs. 0–4; both *p* < 0.001). Discrimination was modest (C-index 0.722) and slightly lower than IMDC (C-index 0.745). NLR ≥ 3.2 was associated with inferior OS (19 vs. 37 months; *p* < 0.001). Bone metastasis was not significantly associated with OS (*p* = 0.27). **Conclusions:** Meet-URO is a valid prognostic tool for mRCC patients treated with first-line TKIs and identifies an ultra-high-risk subgroup (score = 9) with poor survival. While not superior to IMDC, Meet-URO may offer complementary risk information to support clinical monitoring and trial referral, particularly in settings where ICI combinations are restricted.

## 1. Introduction

The use of combination regimens in the treatment of metastatic RCC has gradually increased in recent years and has become the standard treatment in most patients with current developments [1]. After the approval of nivolumab in 2015, immune checkpoint inhibitor therapies came into use, and shortly afterwards, ICI-based combination strategies were found to contribute to overall survival [2,3]. Real-world data show that the use of first-line tyrosine kinase inhibitor monotherapy has declined from 77% to 10–27% since 2020 [4,5]. Despite this decline, TKIs are still an important option in some low-risk patient groups, in developing countries or in countries with insurance-related problems in accessing ICI combinations [6,7].

The International Metastatic RCC Database Consortium (IMDC) score is the most commonly used prognostic scoring system when planning treatment in metastatic RCC. Although it was recommended in the patient group receiving TKIs in 2009, it is now used in almost all patients [8]. Nevertheless, the search continues for a new prognostic scoring system that includes larger patient cohorts and is more inclusive and effective.

The IMDC score is the most commonly applied prognostic model in mRCC and, although developed in TKI-treated populations, it is now used across treatment classes. The Meet-URO score augments IMDC by adding NLR and the presence of bone metastases [9]. The scoring model has been shown to be superior to IMDC in patients receiving first-line nivolumab plus ipilimumab, ICI + TKI combination, cabozantinib in the 2–3rd line, and cabozantinib in patients over 70 years of age [10,11,12,13,14]. Whether Meet-URO provides clinically useful risk stratification among patients receiving first-line TKI monotherapy in routine practice remains uncertain.

We evaluated the prognostic performance of the Meet-URO score in a real-world cohort of patients with mRCC who initiated first-line TKI monotherapy at five tertiary centers.

## 2. Material and Methods

### 2.1. Patient Characteristics and Study Design

We conducted a retrospective, multicenter cohort study across five tertiary-care institutions in Turkey: Dokuz Eylül University (Izmir), Celal Bayar University (Manisa), Hacettepe University (Ankara), Çukurova University (Adana), and Bilkent City Hospital (Ankara). The study adhered to the Declaration of Helsinki and was approved by the Dokuz Eylül University Ethics Committee (Protocol 9663-GOA; decision 2025/16-04).

Eligible patients were ≥18 years old with histologically confirmed mRCC (clear-cell or non-clear-cell) who received first-line TKI monotherapy (sunitinib, pazopanib, or cabozantinib) for ≥28 days between January 2008 and January 2025. Exclusion criteria included prior systemic therapy, combination regimens, death within 28 days of TKI start, missing key baseline data, or loss to follow-up.

All initial treatment decisions were made by a multidisciplinary tumor board (urology and medical oncology; radiation oncology/nuclear medicine as needed). Sunitinib was administered 50 mg daily on a 4-weeks-on/2-weeks-off schedule with permitted dose reductions to 37.5 mg then 25 mg daily. Pazopanib was initiated at 800 mg once daily with stepwise reductions to 400 mg and 200 mg as needed. Cabozantinib was started at 60 mg once daily with permitted reductions to 40 mg and 20 mg. Dosing and supportive care followed European Society for Medical Oncology and European Association of Urology guidance contemporaneous with the treatment period (2008–2025) and local reimbursement policies [15,16].

The primary endpoint was overall survival (OS), defined from first TKI dose (day 0) to death from any cause. Treatment continued until radiologic/clinical progression, unacceptable toxicity, withdrawal, or death. Baseline laboratory values (including complete blood count for NLR) and Karnofsky Performance Status were abstracted from the closest measurement within 14 days before day 0. IMDC and Meet-URO scores were calculated at day 0 using prespecified definitions.

Radiologic assessments (contrast-enhanced CT chest/abdomen/pelvis and, when indicated, head imaging) were performed at baseline and approximately every 12 weeks, or earlier as clinically warranted, and responses were evaluated per RECIST v1.1.

### 2.2. Statistical Analysis

Medians with ranges were used for continuous variables and percentages for categorical data to characterize patient characteristics. Fisher’s exact test or the chi-squared test, depending on the situation, was used to examine relationships between variables. The Meet-URO score was computed using baseline characteristics, including the neutrophil-to-lymphocyte ratio (NLR), the existence of bone metastases, and IMDC risk categories, in order to facilitate risk classification (web calculator address: https://proviso.shinyapps.io/Meet-URO15_score/ (accessed on 1 August 2025). Both progression-free survival (PFS) and overall survival (OS) were estimated using Kaplan–Meier curves. Both univariate and multivariate analyses were performed using Cox proportional hazards models to examine the effects of inflammatory markers, treatment lines, and other clinical variables on outcomes. A two-sided *p*-value of less than 0.05 was deemed statistically significant in the statistical analyses, which were carried out using IBM SPSS Statistics v30 (Windows/Mac) (SPSS Inc., Chicago, IL, USA). Harrell’s C-index, used to evaluate model discrimination, was calculated in R (The R Foundation for Statistical Computing, Vienna, Austria). Harrell’s C-index was estimated with 95% confidence intervals using 1000-bootstrap resampling. Pre-specified risk strata were pragmatically collapsed from the original five tiers to three (0–4, 5–8, 9) to reduce sparse-event uncertainty and improve clinical actionability. Sensitivity analyses with the original five-tier stratification (KM) are provided in the Appendix A

## 3. Results

The study included 301 patients, most of whom were under 70 years old (236 patients, 78.4%), and 89 of whom were female (29.6%). Patients’ characteristics are summarized in Table 1. The majority of patients exhibited a favorable performance status, with 253 (84.1%) demonstrating a Karnofsky score of ≥80%. The predominant subtype, observed in 246 patients (81.7%), was clear cell histology, while non-clear cell histology was present in 55 (18.3%) patients. A prior nephrectomy had been performed in 190 patients (63.1%). De novo metastatic disease at initial diagnosis was documented in 178 patients (59.1%), while 123 (40.9%) exhibited recurrent disease.

A comparison of the Meet-URO prognostic score groups revealed significant differences in Karnofsky performance status, prior nephrectomy, presence of metastasis at diagnosis, IMDC risk classification, presence of bone metastases, and NLR levels. Patients in the lower Meet-URO score group showed higher rates of good performance status (Karnofsky ≥ 80%: 90.6% vs. 64.3%), more frequent nephrectomy (71.9% vs. 32.1%), and increased likelihood of recurrent disease (50.4% vs. 10.7%) compared to those in the highest score group. Conversely, patients with high risk characteristics were more likely to present with de novo metastatic disease, elevated neutrophil-to-lymphocyte ratio (NLR) values greater than or equal to 3.2, and bone metastases in 100% of cases (all *p* < 0.001). The study revealed no statistically significant differences between the groups with respect to sex, age group, histologic subtype, brain and liver metastases, sarcomatoid features, or type of first-line systemic treatment (all *p* > 0.05).

Median follow-up was 40 months (CI: 31–51), median OS (mOS) was 25 (CI: 21–29) months, and median PFS (mPFS) was 10 (CI: 8–12) months. For PFS, 301 patients were evaluable; 269/301 (89.9%) experienced a PFS event (RECIST 1.1 progression or death), and 32/301 (10.6%) were censored. For OS, 301 patients were evaluable; 228/301 (75.7%) died and 73/301 (24.3%) were censored.

Univariate Cox regression analysis was conducted to evaluate the impact of the three prognostic factors that make up the Meet-URO score on survival outcomes. In univariate analysis, the IMDC intermediate-risk group had significantly worse OS than the favorable-risk group (median 30 vs. 48 months; HR 1.96, 95% CI 1.09–3.55; *p* = 0.025). Patients classified as poor-risk had a notably shorter mOS compared to those in the favorable-risk group (11 months versus 48 months, *p* < 0.001) (Figure 1). The discriminative ability (score based) of the IMDC score was reflected in a C-index of 0.714 (95% CI: 0.670–0.758, *p* < 0.001).

An elevated NLR (≥3.2) was significantly associated with shorter median OS: 19 months compared to 37 months in patients with lower NLR (*p* < 0.001) (Figure 2). While the presence of bone metastases was associated with a numerically lower mOS (21 vs. 28 months), this difference did not reach statistical significance (*p* = 0.27) (Table 2) (Figure 3).

Meet-URO score was calculated for all patients and its distribution is shown in Figure 4. The most common score was 3 (26.6%), followed by 5 (16.3%). A Meet-URO score ranging from 0 to 9 showed prognostic value with a Harrell C-index of 0.680 (95% CI: 0.636–0.725, *p* < 0.001).

However, the intersection of the Kaplan–Meier survival curves of groups 2 and 3 in the original five-risk category grouping system necessitated the implementation of ROC analysis, which revealed a more pronounced discrepancy between scores of 4 and 5. Consequently, patients were stratified into two distinct prognostic categories, namely scores of 0–4 and ≥5 (Appendix A). Patients with the highest score of 9 exhibited particularly poor outcomes and were thus placed in a separate category, a methodology consistent with the study by Rebuzzi et al. [17]. Based on these findings, three prognostic groups were formed: group 1 (score 0–4; n = 139, 46.2%), group 2 (score 5–8; n = 134, 44.5%) and group 3 (score 9; n = 28, 9.3%). Median OS differed significantly between groups, with 21 months in group 2 (*p* = 0.001) and 7 months in group 3 (*p* < 0.001) compared to 41 months in group 1 (Figure 5).

Univariate and multivariate Cox regression analyses including the Meet-URO score groups are shown in Table 3.

Compared with the IMDC score groups with a C-index of 0.745 (95% CI: 0.687–0.802, *p* < 0.001), the three-stage classification based on the Meet-URO score showed lower prognostic discrimination with a lower C-index of 0.722 (95% CI: 0.665–0.778, *p* < 0.001).

## 4. Discussion

In this multicenter retrospective analysis, we validated the prognostic utility of the Meet-URO score in patients with metastatic renal cell carcinoma (mRCC) treated with first-line tyrosine kinase inhibitor (TKI) monotherapy. The score effectively stratified risk with statistically significant differences in overall survival (OS), even within a TKI-treated population. Patients with lower Meet-URO scores (0–4) experienced substantially longer OS than those with intermediate (5–8) or high (9) scores, and a score of 9 identified an ultra-high-risk subgroup with a median OS of 7 months.

Originally developed and validated in immune checkpoint inhibitor (ICI) cohorts, the Meet-URO framework integrates systemic inflammation (neutrophil-to-lymphocyte ratio, NLR) and bone metastasis into the IMDC construct [9,18]. Real-world evidence among patients receiving first-line TKIs has been limited; our data fill this gap and show that Meet-URO retains prognostic relevance in this setting.

When compared with IMDC, Meet-URO demonstrated broadly comparable discrimination (C-index 0.722 vs. 0.745). The relatively modest difference may reflect the lack of association between bone metastasis and OS in our cohort. Although bone involvement is often considered an adverse feature in RCC, prior studies have reported heterogeneous effects [19]: Roviello et al. observed no significant relationship between bone metastasis and survival among good-risk patients [20]; a meta-analysis by Bersanelli et al. reported similar survival independent of bone involvement with cabozantinib [21]; and in a first-line TKI setting, Kang et al. found that bone metastasis predicted worse outcomes only in the IMDC intermediate-risk subgroup [22]. These heterogeneous findings in TKI-treated populations raise important questions about the role of bone metastasis within the Meet-URO scoring system in such contexts. Indeed, the lack of superiority of the Meet-URO score over the IMDC model in our study further supports these concerns. The absence of a significant association between bone metastases and OS may reflect sample size, imaging heterogeneity, and the mitigating effect of supportive measures (e.g., radiotherapy, bone-targeted agents). Therefore, we preserve the bone metastasis component and frame our finding as hypothesis-generating. A modified model will require re-estimation and external validation in populations dominated by TKIs. Future studies should investigate the re-calibration and re-calculation of coefficients using contemporary TKI/ICI era datasets.

Higher NLR was strongly and consistently associated with inferior survival in our study, supporting prior evidence that systemic inflammation indexes more aggressive disease biology [23,24]. Although a universally accepted NLR threshold is lacking, prior work has suggested a cut-off around 3.2 [9], which separated outcomes in our cohort.

The IMDC model also performed well, stratifying patients into favorable-, intermediate-, and poor-risk groups with statistically significant OS differences; notably, the separation between favorable and intermediate risk (48 vs. 30 months; *p* = 0.025) was clinically meaningful. External series reinforce IMDC’s robustness in TKI-treated practice, including a multicenter cohort from Spain [25] and a report by Sobu et al. showing associations between IMDC poor risk, TKI choice, and outcomes [26].

In our study, the IMDC model was less effective in distinguishing ultra-poor prognosis patients (OS < 10 months), a subgroup that was better identified using the Meet-URO score. The IMDC score showed strong discriminatory power (C-index: 0.745) and was superior to the Meet-URO score in the whole population. This may be because the presence of bone metastases has no significant effect on survival. Another reason may be that the IMDC score was generated from a cohort of patients receiving TKI-based therapy, which may have better predicted the survival of patients receiving first-line TKI therapy [8]. The REGAL study, once finalized, is expected to provide a substantial contribution to the literature because it includes the patient group receiving first-line TKIs [27].

Implications for practice are twofold. First, reporting Meet-URO alongside IMDC may help flag ultra-high-risk patients (score = 9) who warrant earlier treatment reassessment or escalation, closer toxicity–response surveillance, proactive supportive/palliative care, and timely trial referral. Second, because Meet-URO relies on routinely available parameters (CBC-derived NLR and standard clinical variables), it is feasible in resource-constrained settings and can support risk-adapted care and trial stratification when access to ICI combinations is limited.

However, while the IMDC score remains a practical and accessible tool, its limitations warrant consideration. It does not take into account emerging prognostic variables such as site-specific metastases (e.g., bone, liver), molecular or immunohistochemical markers [28,29]. Nevertheless, the inclusion of additional biological parameters in the models may improve risk stratification and clinical decision-making, especially in complex real-world populations.

Despite its strengths, our study has limitations. Its retrospective, multicenter design across five tertiary centers may introduce selection and information bias and practice heterogeneity. Baseline variables for IMDC and Meet-URO were abstracted from measurements within 14 days before treatment start, potentially adding variability to NLR and performance status. The use of three different TKIs with permitted dose modifications reflects real world practice but raises confounding by indication and dosing; exclusion of deaths within 28 days of initiation may bias OS by removing early events. We evaluated Meet-URO without recalibration for a TKI treated population and collapsed five strata to three, which may entail some information loss and limit formal performance characterization; estimates for the ultra-high risk tier (score = 9) are imprecise due to small numbers. Some components, including bone metastasis, did not achieve statistical significance, which may reflect limited power and heterogeneity rather than absence of effect. Generalizability is limited to first-line TKI monotherapy in a single-country setting during a period when access to ICI combinations and reimbursement policies were evolving. In addition, surgical pathology covariates (e.g., margin status, lymphadenectomy) and patient-reported outcomes were not systematically captured, and external validation was not performed.

This multicenter study shows that the Meet-URO score is a valid and practical prognostic tool for mRCC patients receiving first-line TKI monotherapy. Developed for immunotherapy populations, the Meet-URO score maintains prognostic relevance in the TKI setting and identifies patients with poor survival. While it does not outperform the IMDC score, it provides complementary value by identifying ultra-high-risk patients who may benefit from more intensive monitoring or novel approaches. These findings support using the Meet-URO score as an adjunct to established models like IMDC and highlight the need for validation in future studies, including those using immunotherapy and biomarker-driven strategies. Meet-URO retains prognostic value in patients with mRCC receiving first line TKI monotherapy and complements IMDC by identifying an ultra-high risk subgroup; we therefore recommend concurrent reporting of Meet-URO with IMDC to support risk adapted clinical decision making and trial stratification, alongside prospective, centrally reviewed, and externally validated studies with standardized baseline capture and potential recalibration of Meet-URO for TKI dominant and combination eras.

## Figures and Tables

**Figure 1 jcm-14-06385-f001:**
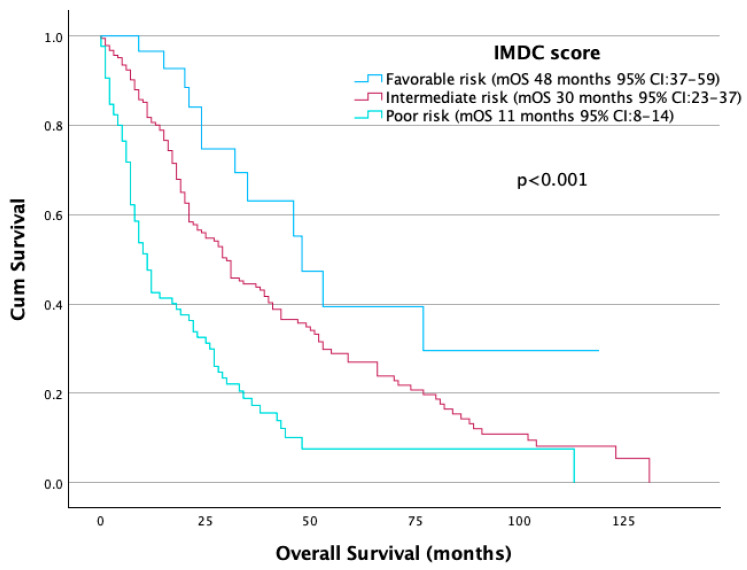
OS according to IMDC score.

**Figure 2 jcm-14-06385-f002:**
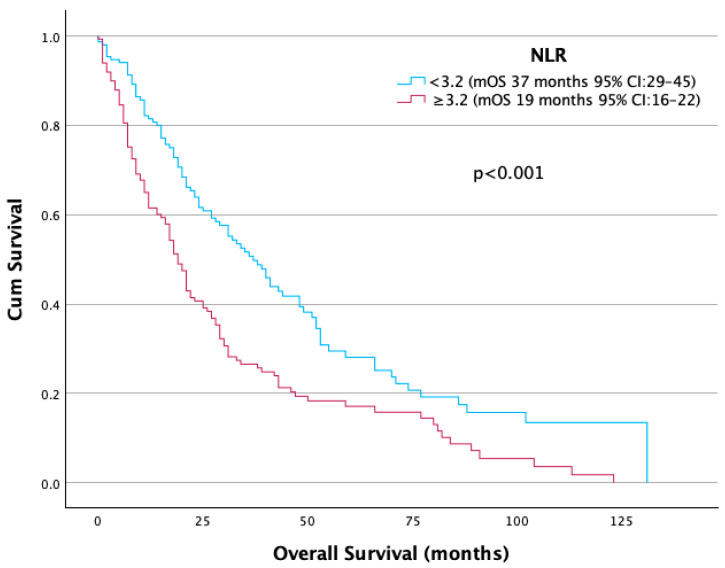
OS according to NLR.

**Figure 3 jcm-14-06385-f003:**
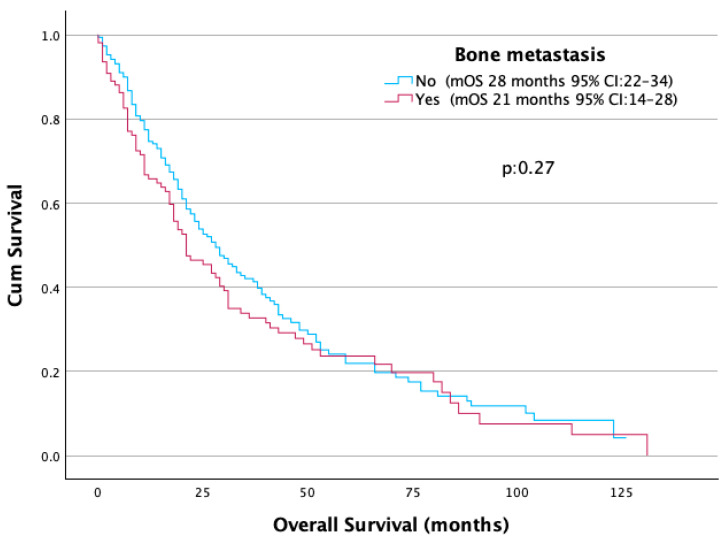
OS according to presence of bone metastasis.

**Figure 4 jcm-14-06385-f004:**
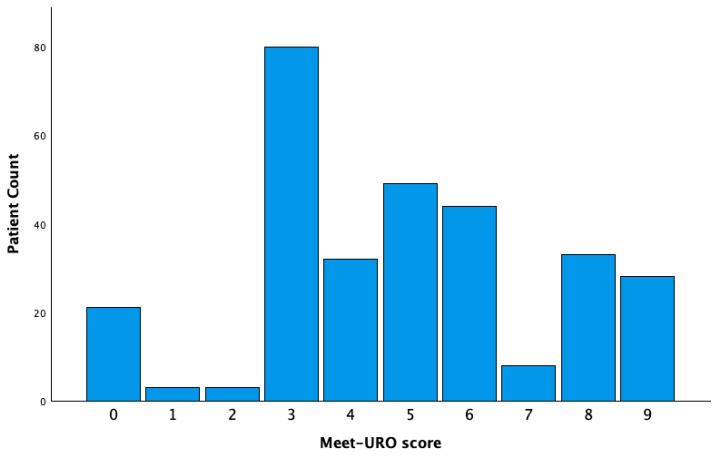
Patient distribution according to Meet-URO score.

**Figure 5 jcm-14-06385-f005:**
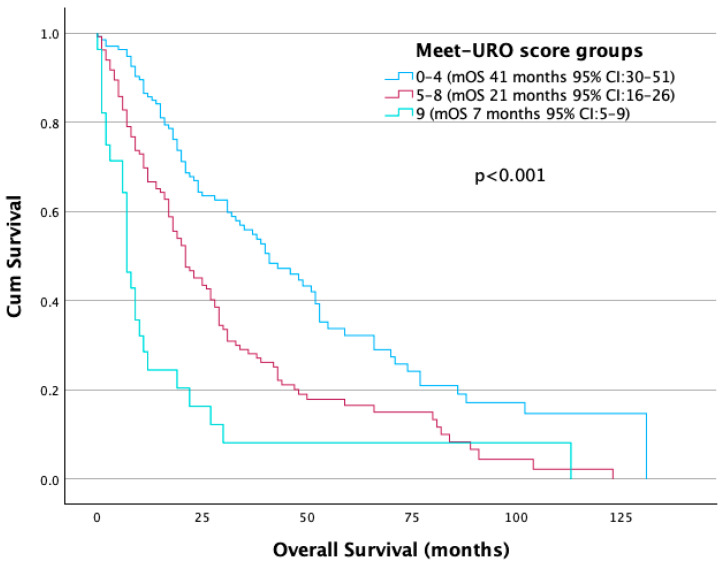
OS according to Meet-URO score.

**Table 1 jcm-14-06385-t001:** Patient Characteristics.

	Meet-URO Score Prognostic Group				
Characteristics	All Cohort (*n* = 301, 100%)	0–4 (*n* = 139, 46.2%)	5–8 (*n* = 134, 44.5%)	9 (*n* = 28, 9.3%)	*p* *-Value
Sex					
Female	89 (29.6)	37 (26.6)	43 (32.1)	9 (32.1)	
Male	212 (70.4)	102 (73.4)	91 (67.9)	19 (67.9)	0.58
Age group					
<70	236 (78.4)	114 (82.0)	99 (73.9)	23 (82.1)	
≥70	65 (21.6)	25 (18.0)	35 (26.1)	5 (17.9)	0.23
Karnofsky PS					
<80%	48 (15.9)	13 (9.4)	25 (18.7)	10 (35.7)	
≥80%	253 (84.1)	126 (90.6)	109 (81.3)	18 (64.3)	0.001
Histologic subtype					
Clear cell	246 (81.7)	117 (84.2)	109 (81.3)	20 (71.4)	
Non-clear cell	55 (18.3)	22 (15.8)	25 (18.7)	8 (28.6)	0.28
Nephrectomy					
No	111 (36.9)	39 (28.1)	53 (39.6)	19 (67.9)	
Yes	190 (63.1)	100 (71.9)	81 (60.4)	9 (32.1)	<0.001
Metastatic at diagnosis					
Recurrent	123 (40.9)	70 (50.4)	50 (37.3)	3 (10.7)	
De-novo	178 (59.1)	69 (49.6)	84 (62.7)	25 (89.3)	<0.001
IMDC risk group					
Good	31 (10.3)	31 (22.3)	0 (0.0)	0 (0.0)	
Intermediate	185 (61.5)	108 (77.7)	77 (57.5)	0 (0.0)	
Poor	85 (28.2)	0 (0.0)	57 (42.5)	28 (100.0)	<0.001
Bone metastasis					
No	191 (63.5)	96 (69.1)	95 (70.5)	0 (0.0)	
Yes	110 (36.5)	43 (30.9)	39 (29.1)	28 (100.0)	<0.001
Brain metastasis					
No	277 (92.0)	125 (89.9)	125 (93.3)	27 (96.4)	
Yes	24 (8.0)	14 (10.1)	9 (6.7)	1 (3.6)	0.39
Liver metastasis					
No	236 (78.4)	109 (78.4)	106 (79.1)	21 (75.0)	
Yes	65 (21.6)	30 (21.6)	28 (20.9)	7 (25.0)	0.89
Sarcomatoid differentiation					
No	284 (94.4)	135 (97.1)	123 (91.8)	26 (92.9)	
Yes	17 (5.6)	4 (2.9)	11 (8.2)	2 (7.1)	0.15
NLR					
<3.2	152 (50.5)	128 (92.1)	24 (17.9)	0 (0.0)	
≥3.2	149 (49.5)	11 (7.9)	110 (82.1)	28 (100)	<0.001
First-Line Treatment					
Sunitinib	145 (48.2)	76 (54.7)	56 (41.8)	13 (46.4)	
Pazopanib	125 (41.5)	53 (38.1)	60 (44.8)	12 (42.9)	
Cabozantinib	31 (10.3)	10 (7.2)	18 (13.4)	3 (10.7)	0.22

* <0.05.

**Table 2 jcm-14-06385-t002:** Univariate analyses of the prognostic factors that contribute to the Meet-URO score.

Prognostic Factors	Values	mOS	HR	*p*-Value
IMDC score	Favorable	48	1	<0.001
	Intermediate	30	1.96 (1.09–3.55)	0.025
	Poor	11	4.37 (2.37–8.07)	<0.001
NLR	<3.2	37	1	
	≥3.2	19	1.77 (1.35–2.31)	<0.001
Bone metastasis	Yes	21	1.17 (0.89–1.53)	0.27
	No	28	1	

**Table 3 jcm-14-06385-t003:** Univariate and multivariate analysis for OS.

	Univariate Analysis		Multivariate Analysis	
	HR (95% CI)	*p*-Value *	HR (95% CI)	*p*-Value *
Age (<70 vs. ≥70)	0.99 (0.72–1.38)	0.98		
Gender (woman vs. man)	0.83 (0.63–1.11)	0.21		
De-novo vs. relapse	1.27 (0.97–1.66)	0.08		
Meet-URO risk group		<0.001		<0.001
(1 vs. 2)	1.87 (1.4–2.48)	<0.001	1.73 (1.29–2.32)	<0.001
(1 vs. 3)	4.0 (2.58–6.31)	<0.001	3.57 (2.25–5.65)	<0.001
Nephrectomy (yes or no)	0.52 (0.41–0.71)	<0.001	0.59 (0.45–0.8)	<0.001
Histology (clear cell vs. non clear cell)	1.12 (0.79–1.59)	0.51		
First-line treatment		0.057		0.17
(sunitinib vs. pazopanib)	1.24 (0.94–1.64)	0.13	1.15 (0.87–1.5)	0.33
(sunitinib vs. cabozantinib)	1.86 (1.06–3.24)	0.03	1.7 (0.96–2.98)	0.07

* <0.05.

## Data Availability

The data that support the findings of this study are available on request from the corresponding author.

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
