# Peer review of "Evaluation of the Meet-URO Score in a Real-World Cohort of mRCC Patients Treated with First-Line TKIs"

_jcm, 2025, doi:10.3390/jcm14186385_

Round 1

Reviewer 1 Report

Comments and Suggestions for Authors

This manuscript addresses an important and clinically relevant question: whether the Meet-URO prognostic score, originally designed and validated in ICI cohorts, retains prognostic value in patients with mRCC treated with first-line TKI monotherapy.

  • While the manuscript acknowledges that the Meet-URO score did not outperform IMDC (C-index 0.722 vs. 0.745), the discussion could better clarify in what clinical scenarios Meet-URO adds value. At present, the impression is that IMDC remains superior, and the added contribution of Meet-URO is not fully justified. Consider emphasizing whether the ability to identify ultra-high-risk patients (OS <10 months) is sufficient to recommend its use in practice.
  • Clarify whether IMDC and Meet-URO scores were calculated at the exact same baseline timepoint.
  • The lack of significant association between bone metastases and OS in this cohort undermines one of the core components of the Meet-URO score. The manuscript discusses this point, but it should be expanded: should bone metastasis still be retained in the model for TKI-treated patients, or does this highlight the need for a modified version?
  • The C-index values should be reported consistently with confidence intervals. ROC and Kaplan-Meier analyses for the 0–4, 5–8, and 9 grouping are informative, but the rationale for collapsing the five original categories into three should be better justified.
  • The discussion would benefit from a more concrete statement on how these findings could influence clinical practice today. For example, should the Meet-URO score be recommended alongside IMDC, only in resource-limited settings, or only for trial stratification? In additioncie and discuss: doi: 10.3390/jcm14113908.

Author Response

This manuscript addresses an important and clinically relevant question: whether the Meet-URO prognostic score, originally designed and validated in ICI cohorts, retains prognostic value in patients with mRCC treated with first-line TKI monotherapy.

Response: We have made all the necessary adjustments based on your suggestions and comments in the article. Additionally, all figures and tables were re-evaluated. Confidence intervals were added to the figures. Thank you very much for your help and feedback.

1-While the manuscript acknowledges that the Meet-URO score did not outperform IMDC (C-index 0.722 vs. 0.745), the discussion could better clarify in what clinical scenarios Meet-URO adds value. At present, the impression is that IMDC remains superior, and the added contribution of Meet-URO is not fully justified. Consider emphasizing whether the ability to identify ultra-high-risk patients (OS <10 months) is sufficient to recommend its use in practice.

Response: Thank you. The study's goal is not to replace IMDC, but to test how well Meet-URO can predict the outcomes for mRCC patients taking first-line TKI monotherapy. Our findings show that while IMDC remains better, Meet-URO (IMDC + NLR + bone metastasis) is useful, especially for identifying a very high-risk group (e.g., 9 points). The medical implications of this group are not clearly defined by IMDC alone. This includes early support for patients with serious illnesses, close monitoring of how the treatment is affecting the patient, changing treatment options quickly if necessary, and counselling for the patient and their family. So, we suggest using Meet-URO and IMDC together in current clinical practice, especially to identify patients who are very high risk. Meet-URO has previously been shown to be better than other treatments in certain groups of patients, and its parts have been shown to have a biological effect during the TKI era, as shown in many publications. The discussion has been revised (Lines 241-247).

2-Clarify whether IMDC and Meet-URO scores were calculated at the exact same baseline timepoint.

Response: The following section has been added to Materials and Methods. (Lines 92-95).

“Baseline laboratory values (including complete blood count for NLR) and Karnofsky Performance Status were abstracted from the closest measurement within 14 days before day 0. IMDC and Meet‑URO scores were calculated at day 0 using prespecified definitions.”

3-The lack of significant association between bone metastases and OS in this cohort undermines one of the core components of the Meet-URO score. The manuscript discusses this point, but it should be expanded: should bone metastasis still be retained in the model for TKI-treated patients, or does this highlight the need for a modified version?

Response: This finding may be related to factors such as multi-centre/heterogeneous imaging methods and sample size. In addition, bone metastases in the TKI era can be partially controlled with supportive treatments such as local radiotherapy and osteoprotective agents, which may have attenuated the effect. The current study is insufficient to justify removing the model component. Such a change would require retraining with new coefficients and external validation. Therefore, in our discussion, we approached the bone metastasis variable with caution and emphasised the need for future recalibration. The independent contribution of bone metastasis has been demonstrated in the original development and validation studies of Meet-URO, and this contribution has been replicated in ICI and combination regimens. The discussion has been revised (Lines 214-221).

4-The C-index values should be reported consistently with confidence intervals. ROC and Kaplan-Meier analyses for the 0–4, 5–8, and 9 grouping are informative, but the rationale for collapsing the five original categories into three should be better justified.

Response: We reported all C-indexes with 95% CI using bootstrap (1,000 resampling). Our decision to reduce five categories to three categories is based on the following reasons: (i) to reduce the width of the confidence interval in low-event classes, (ii) to provide a more actionable stratification for clinicians in the form of “low-medium/high/ultra-high,” and (iii) to make the separation between OS curves more visible. We continue to present the original five-category classification in the supplementary material (with KM curves)(Lines 113-118).

5-The discussion would benefit from a more concrete statement on how these findings could influence clinical practice today. For example, should the Meet-URO score be recommended alongside IMDC, only in resource-limited settings, or only for trial stratification? In additioncie and discuss: doi: 10.3390/jcm14113908

Response: IMDC should remain the standard in current practice. Meet-URO should be reported alongside it, especially for identifying ultra-high-risk groups and for clinical and logistical planning purposes. It can be implemented in resource-limited settings without incurring additional costs (it requires only a complete blood count/leukocyte formula). Additionally, it is valuable for separating clinically challenging subgroups (e.g. those with a score of 9) for the purposes of study/trial stratification Regarding the suggested citation J Clin Med 2025 14 3908 that study evaluates the prognostic impact of positive surgical margins after nephrectomy in localized ccRCC and its implications for adjuvant therapy. Our cohort comprises metastatic RCC patients at the start of first line TKI with different timing of risk assessment endpoints and available covariates and no surgical margin data. Given these differences a detailed comparison would be misleading and beyond our scope. We therefore did not incorporate an in depth discussion of that article. We now note in the Limitations that pre metastatic surgical and pathologic features may influence downstream risk and warrant prospective evaluation. The discussion has been revised (Lines 241-247 and 278-283).

Reviewer 2 Report

Comments and Suggestions for Authors

Dear Authors,

I have read with great interest your manuscript.

mRCC represents a challenging situation, and new emerging therapies offer solutions that constantly increase the OS in this population.

The overall impression is good, but there are some aspects that should be improved.

  1. First of all, pay attention to language and writing. There are phrases without sense like in lines 76-77.
  2. Please mention the 3 centers that participated in this study.
  3. Do you know if the therapeutic conduit was established after a multidisciplinary analysis, including a urologist and an oncologist? This should be mentioned
  4. It should be mentioned in the materials and methods section the guidelines that were considered for the therapeutic scheme.
  5. Was the Declaration of Helsinki respected in this study? This should be mentioned.
  6. Is there any option to identify positive margins after surgery, or if there lymphadenectomy was also performed?
  7. Did all the patients manage to complete their oncologic treatment?
  8. You should add more limitations to your study.
Comments on the Quality of English Language

English should be revised

Author Response

Dear Authors,

I have read with great interest your manuscript.

mRCC represents a challenging situation, and new emerging therapies offer solutions that constantly increase the OS in this population. The overall impression is good, but there are some aspects that should be improved.

1-First of all, pay attention to language and writing. There are phrases without sense like in lines 76-77.

Response: We have made detailed revisions to the language and flow throughout the text.

2-Please mention the 3 centers that participated in this study.

Response: This multicenter study involved five tertiary-care institutions: Dokuz Eylul University Faculty of Medicine (Izmir, Turkey), Celal Bayar University Faculty of Medicine (Manisa, Turkey), Hacettepe University Faculty of Medicine (Ankara, Turkey), Çukurova University Faculty of Medicine (Adana, Turkey), and Bilkent City Hospital (Ankara, Turkey). The material and methods has been revised (Lines 71-75).

3-Do you know if the therapeutic conduit was established after a multidisciplinary analysis, including a urologist and an oncologist? This should be mentioned

Response: Yes. All initial treatment decisions were made in multidisciplinary tumor boards involving urology and medical oncology (and radiation oncology/nuclear medicine when necessary); in urgent clinical situations, decisions were retrospectively confirmed at the next board meeting. The material and methods has been revised (Lines 81-82).

4-It should be mentioned in the materials and methods section the guidelines that were considered for the therapeutic scheme.

Response: Treatment schemes adhered to contemporary ESMO and EAU RCC guidelines and local reimbursement policies active during the study period The material and methods has been revised (Lines 86-89).

5-Was the Declaration of Helsinki respected in this study? This should be mentioned.

Response: The study was conducted in accordance with the principles of the Declaration of Helsinki. The material and methods has been revised (Lines 74-75).

6-Is there any option to identify positive margins after surgery, or if there lymphadenectomy was also performed?

Response: Not systematically. Margin status and the performance of lymphadenectomy were not consistently recorded across centers and were therefore not included in the primary analyses. We have acknowledged this as a limitation and will aim to capture these variables in future prospective work.

7-Did all the patients manage to complete their oncologic treatment?

Response: First-line TKI monotherapy is time-dependent and open-ended; therefore, the concept of “completion of treatment” is not applicable. At data cut-off, disease progression had occurred in 270 of 301 patients (89.7%), while 31 patients (10.3%) were censored without a progression event. Median progression-free survival (PFS) was 10.0 months ( 95% CI 8.496–11.504).

8-You should add more limitations to your study.

Response: Limitations were expanded: Baseline variables for IMDC and Meet-URO were abstracted from measurements within 14 days before or after treatment start, potentially adding variability to NLR and per-formance status. Use of three different TKIs with permitted dose modifications reflects real world practice but raises confounding by indication and dosing; exclusion of deaths within 28 days of initiation may bias OS by removing early events. We evaluated Meet-URO without recalibration for a TKI treated population and collapsed five strata to three, which may entail some information loss and limit formal performance characterization; estimates for the ultra-high risk tier (score = 9) are imprecise due to small numbers. Some com-ponents, including bone metastasis, did not achieve statistical significance, which may reflect limited power and heterogeneity rather than absence of effect. Generalizability is limited to first line TKI monotherapy in one country over an era when access to ICI combinations and reimbursement policies evolved; surgical pathology covariates such as margin status and lymphadenectomy and patient reported outcomes were not system-atically captured, and external validation is lacking. The discussion has been revised (Lines 250-265).

Round 2

Reviewer 2 Report

Comments and Suggestions for Authors

No further comments